# DATA-AWARE LOW-RANK COMPRESSION FOR LARGE NLP MODELS

## ABSTRACT

The representations learned by large-scale NLP models such as BERT have been widely used in various tasks. However, the increasing model size of the pre-trained models also brings the efficiency challenges, including the inference speed and the model size when deploying the model on devices. Specifically, most operations in BERT consist of matrix multiplications. These matrices are not low-rank and thus canonical matrix decomposition could not find an efficient approximation. In this paper, we observe that the learned representation of each layer lies in a low-dimensional space. Based on this observation, we propose DRONE (**d**ata-awa**r**e l**o**w-ra**n**k compr**e**ssion), a provably optimal low-rank decomposition of weight matrices, which has a simple closed form solution that can be efficiently computed. DRONE is generic, could be applied to both fully-connected and self-attention layers, and does not require any fine-tuning or distillation steps. Experimental results show that DRONE could improve both model size and inference speed with limited loss of accuracy. Specifically, DRONE alone achieves 1.92x faster on MRPC task with only 1.5% loss of accuracy, and when combined with distillation, DRONE achieves over 12.3x faster on various natural language inference tasks.

## 1 INTRODUCTION

The representations learned by large-scale Natural Language Processing (NLP) models such as BERT have been widely used in various tasks (Devlin et al., 2018). The pre-trained models of BERT and its variations are used as feature extractors for the downstream tasks such as question answering and natural language understanding (Radford et al.; Howard & Ruder, 2018). The success of the pre-trained BERT relies on the usage of large corpus and big models. Indeed, researchers have reported better results of models with more parameters (Shazeer et al., 2018) and number of layers (Al-Rfou et al., 2019). The increasing model size of the pre-trained models inhibits the public user from training a model from scratch, and it also brings the efficiency challenges, including the inference speed and the model size when deploying the model on devices.

To deal with the efficiency issue, most existing works resort to adjusting the model structures or distillation. For instance, Kitaev et al. (2020) uses locality-sensitive hashing to accelerate dot-product attention, Lan et al. (2019) uses repeating model parameters to reduce the size and Zhang et al. (2018) applies a pre-defined attention pattern to save computation. A large body of prior work focuses on variants of distillation information has also been explored (Sanh et al., 2019; Jiao et al., 2019; Sun et al., 2020; Liu et al., 2020; Xu et al., 2020; Sun et al., 2019). However, all these methods either require a specific design of model architecture which is not generic, or require users to train the proposed structure from scratch which greatly reduces its practicality.

In this work, we try to explore an acceleration method that is more generic. Note that as shown in Figure 1, matrix multiplication (Feed-forward layer) is a fundamental operation which appears many times in the Transformer architecture. In fact, the underlying computation of both multi-head attention layers and feed-forward layers is matrix multiplication. Therefore, instead of resorting to the complex architecture redesign approaches, we aim to investigate whether low-rank matrix approximation, the most classical and simple model compression approach, can be used to accelerate Transformers. Despite being successfully applied to CNN (Yu et al., 2017; Sindhwani et al., 2015; Shim et al., 2017; You et al., 2019), at the first glance low-rank compression cannot work for BERT. We could see in Figure 2 that regardless of layers, matrices in feed-forward layer, query and key transformation of attention layer **are not low-rank**. Therefore, even the optimal low-rank approximation (e.g., by

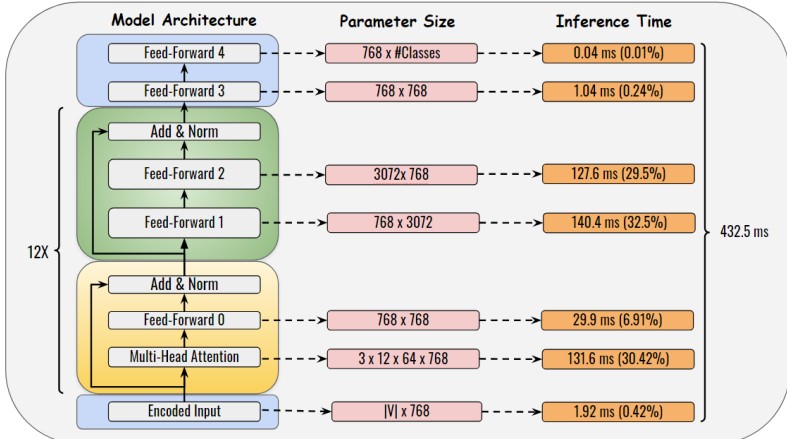

Figure 1: Illustration of the BERT-base computational model. $|V|$ denotes the number of tokens in the model. #Classes denotes the number of classes in the down-stream classification task. Input encoding, Feed-forward 3 and Feed-forward 4 are computed only once in the inference and thus do not contribute to overall computational time much.

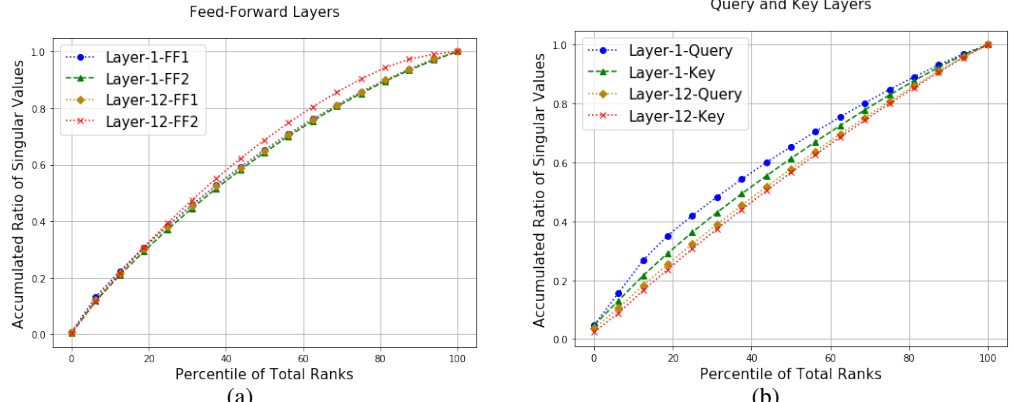

Figure 2: Illustration of the empirical observation that weight matrices in BERT model are not low-rank. The x-axis represents what percentage of the ranks are selected; the y-axis represents sum of singular values connected to the selected ranks divided by sum of all singular values. Ideally, a low-rank structure will have a larger area under curve, meaning that a small percentage of the ranks could explain most of singular values. We observe that the sum of top 50% of the ranks only account about 60% of all singular values for matrices in the BERT model. This shows that the matrices do not have a clear low-rank structure.

SVD) will lead to large reconstruction error and empirically the performance is limited. This is probably why low-rank approximation has not been used in BERT compression.

In this paper, we propose a novel low-rank approximation algorithm to compress the weight matrices even though they are not low-rank. The main idea is to **exploit the data distribution**. In NLP applications, the latent features, indicating some information extracted from natural sentences, often lie in a subspace with a lower intrinsic dimension. Therefore, in most of the matrix-vector products, even though the weight matrices are not low-rank, the input vectors lie in a low-dimensional subspace, allowing dimension reduction with minimal degraded performance. We mathematically formulate this generalized low-rank approximation problem which includes the data distribution term and provide a closed-form solution for the optimal rank-$k$ decomposition of the weight matrices. We propose DRONE method based on this novel **D**ata-awa**R**e l**O**w-ra**N**k compr**E**ssion idea. Our decomposition significantly outperforms SVD under the same rank constraint, and can successfully accelerate the BERT model without sacrificing too much test performance.

## 2 RELATED WORK

The inference speed is important for NLP models when deployed in various applications. Generally speaking, inference efficiency could be enhanced by hardware (Shawahna et al., 2018) or lower-level instruction optimization (Ning, 2020). On the other hand, the main focus of the current research is on using algorithmic methods to reduce the computational complexity. It could be mainly categorized into two aspects: Attention Complexity Reduction and Model Size Reduction.

**Attention Complexity Reduction**

Attention mechanism is the building block of transformer model and it attracts most attentions of researcher recently in NLP field (Vaswani et al., 2017). Pre-training on large courpus of BERT, a transformer-based model, has contributed to state-of-the-art performance on various tasks after fine-tuning (Devlin et al., 2018). Attention on sequences of length $L$ is $O(L^2)$ in both computational and memory complexity. This would take long inference time when the sequence is long. Thus, researchers have focused on reducing the complexity of the attention module. Kitaev et al. (2020) uses uses locality-sensitive hashing to reduce the complexity to $O(LlogL)$. Zhang et al. (2018); Child et al. (2019) pre-defined an attention map to have a constant computational time. Goyal et al. progressively eliminates the redundant context vectors within the attended sequence to improve the efficiency of attention in last few layers of the model. Wang et al. (2020) proposes to train the low-rank attention by choosing a rank $r \ll L$. This is similar with our work in the sense of leveraging low-rank structures. But our method do not require retraining the model and could be applied to different modules other than attention. In fact, most of these methods require special modules and thus we need to retrain the proposed models from scratch. This prohibits the usage of a large body of publicly available open models for faster research progress. More importantly, these methods only focus on the long sequence scenario. We found out that attention module is actually not the bottleneck of inference time in common usage as shown in Figure 1. In most if not all models of common usages, two layers of large feed-forward layer are appended after attention module which incurs much more computational time. Attention complexity reduction only works when a large sequence is used but in current practice this is unusual. Thus, accelerating attention module itself is not contributing to a significant reduction of overall inference time.

**Model Size Reduction**

Efficiency of inference time is also related to model compression. In principle, smaller models would lead to smaller number of operations and thus faster inference time. Sanh et al. (2020) has explored pruning methods on BERT models to eliminate the redundant links, and there is a line of research on pruning methods (Han et al., 2015a;b; Chen et al., 2020). Quantization methods (Zafrir et al., 2019; Hubara et al., 2016; Lin et al., 2016) could convert the 32 bits float models into lower-bits fixed-point representation, and theoretically make model prediction faster with fixed point accelerator. Lan et al. (2019) reduces the model size by sharing of encoder parameters. A large body of prior work focuses on variants of knowledge distillation (Sanh et al., 2019; Jiao et al., 2019; Sun et al., 2020; Liu et al., 2020; Xu et al., 2020; Sun et al., 2019; 2020). These methods use different strategies to distill information from teacher network and reduce the number of layers (Sanh et al., 2019) or hidden dimension size (Jiao et al., 2019). Further, A hybrid compression method by combining matrix factorization, pruning and knowledge distillation is proposed by Mao et al. (2020). Among the above mentioned methods, Quantization requires hardware accelerator to reduce the inference time which is not applicable to general scenario. Pruning methods could only reduce the model size, but the inference speed might not be reduced due to the limitation of sparse operations. Only algorithmic method such as distillation could serve as a generic inference time accelerating method. We want to emphasize that our method is orthogonal to these distillation methods. In fact, the proposed method is a generic acceleration method applicable to all components in most NLP models. In section 4, we show that DRONE can be combined with the distilled models to further improve the performance.

## 3 METHODS

We now introduce a generic algorithm for improving efficiency of matrix multiplication. The computation of Feed-Forward (FF) Layer in the attention models can be described as:

$$h = Wx + b, \tag{1}$$

where $W \in \mathbb{R}^{d_2 \times d_1}$ and $b \in \mathbb{R}^{d_2}$ are model parameters, $x \in \mathbb{R}^{d_1}$ is the latent representation of a token, and $h \in \mathbb{R}^{d_2}$ is the output. Assume the sequence length is $L$, all the token representations $x_1, \ldots, x_L \in \mathbb{R}^{d_1}$ will pass through this same operation, so in practice the whole FF layer can be computed by a matrix-matrix product $W[x_1, \ldots x_n] + b$, and the computation of bias term $b$ would be broadcasted to all $L$ input tokens. In practice we will normally have $L \ll \max(d_1, d_2)$ (e.g., $L = 128$, $d_2 = 3,072$).

A standard way to accelerate the computation is to perform low-rank approximation over W. A low-rank approximation can be acquired by using singular value decomposition (SVD), which achieves the best rank-k approximation in terms of Frobenius norm and we could write $W$ as:

$$W = USV^T \approx U_{W,k} V_{W,k}{}^T,$$

with unitary matrices $U \in \mathbb{R}^{d_2 \times d_2}$, $V \in \mathbb{R}^{d_1 \times d_1}$ and a diagonal matrix $S \in \mathbb{R}^{d_2 \times d_1}$. $U_{W,k} \in \mathbb{R}^{d_2 \times k}$ and $V_{W,k} \in \mathbb{R}^{d_1 \times k}$ are the rank-$k$ approximation matrices by taking $U_{W,k} = U S_k^{\frac{1}{2}}$, $V_{W,k} = S_k^{\frac{1}{2}} V$, where $S_k^{\frac{1}{2}}$ is the square-root of the first $k$ entries of the diagonal matrix $S$. Given such approximation, we could simplify the computation in equation 1 by writing it as:

$$h = Wx + b \approx U_{W,k} V_{W,k}{}^T x + b.$$

After the rank-$k$ low-rank approximation, the computational complexity reduces from $O(d_2 d_1)$ to $O((d_1 + d_2)k)$. When $k$ is small enough, low-rank approximation could not only accelerate the computation (Shim et al., 2017) but also compress the model size (Sainath et al., 2013). However, as we showed in Figure 2, matrices in FF layer of BERT models do not show obvious low-rank structures. Ideally, we want a small percentage of the ranks which containing all large singular values such that sum of singular values connected to the selected ranks divided by sum of all singular values is large. But we could observe that choosing rank $k$ to be larger than 50% of the ranks (e.g., about 0.5 times $\min(d_1, d_2)$) could only accumulate 60 percent of the total singular values. This will lead to a large approximation error. In the meantime, the complexity is still about $O(d_2 d_1)$ and there is no enhancement of speed.

Despite the matrices in the model are not low-rank, here we provide an illustrative example to show that a low-rank computation could still exist when data distribution lies in a lower intrinsic dimension. Suppose we have a $W$ defined as below and the input $x$ lies in a subspace:

$$W = \begin{bmatrix} 7 & 0 & 2 & 3 & 1 \\ 9 & 6 & 7 & 5 & 0 \\ 6 & 1 & 8 & 0 & 3 \\ 4 & 3 & 2 & 1 & 4 \\ 1 & 2 & 2 & 1 & 2 \end{bmatrix}, \quad x \in \text{span}(\begin{bmatrix} 2 \\ 2 \\ 5 \\ 5 \\ 4 \end{bmatrix}, \begin{bmatrix} 1 \\ 1 \\ 2 \\ 2 \\ 6 \end{bmatrix}),$$

In this case, $W$ is a full-rank matrix so there won't be a lossless low-rank approximation on $W$. On the other hand, input data $x$ lies in a 2-dimensional subspace such that we could construct the following computation:

$$\underbrace{\begin{bmatrix} 7 & 0 & 2 & 3 & 1 \\ 9 & 6 & 7 & 5 & 0 \\ 6 & 1 & 8 & 0 & 3 \\ 4 & 3 & 2 & 1 & 4 \\ 1 & 2 & 2 & 1 & 2 \end{bmatrix}}_{W} \underbrace{\begin{bmatrix} 2 & 1 \\ 2 & 1 \\ 5 & 2 \\ 5 & 2 \\ 4 & 6 \end{bmatrix}}_{x} \begin{bmatrix} a \\ b \end{bmatrix} = \underbrace{\begin{bmatrix} 43 & 23 \\ 90 & 39 \\ 66 & 41 \\ 45 & 37 \\ 29 & 21 \end{bmatrix}}_{U} \underbrace{\begin{bmatrix} -1 & -1 & 0.5 & 0.5 & 0 \\ -0.5 & 0 & 0 & 0 & 0.25 \end{bmatrix}}_{V^T} \underbrace{\begin{bmatrix} 2 & 1 \\ 2 & 1 \\ 5 & 2 \\ 5 & 2 \\ 4 & 6 \end{bmatrix}}_{x} \begin{bmatrix} a \\ b \end{bmatrix},$$

which gives a rank-2 matrix $UV^T$ where $W \neq UV^T$ but $Wx = UV^T x$ for any $x$ in the low dimensional space. This shows that even if we can't approximate the $W$ matrix, it is still possible to construct a good low-rank decomposition, and the key will be to exploit the space of input vectors.

### 3.1 DRONE: DATA-AWARE LOW-RANK COMPRESSION

Assume the input $x$ of the FF Layer follows some distribution, instead of minimizing approximation error of weight matrix (for which SVD is optimal), we want to minimize the approximation error of the outputs. Denote $X$ as the $\mathbb{R}^{d_1 \times n}$ matrix where columns of $X$ capture the empirical distribution of

input, our goal is to find a pair of projection matrix $V_{X,k} \in \mathbb{R}^{d_1 \times k}$ and recovery matrix $U_{X,k} \in \mathbb{R}^{d_2 \times k}$ such that the output result is well approximated, and we could rewrite equation 1 as:

$$h = WX + b \approx WU_{x,k}V_{x,k}{}^T X + b = (WU_{x,k})V_{x,k}{}^T X + b = W_{x,k}V_{x,k}{}^T X + b,$$

where $W_{X,k} = WU_{x,k}$. Intuitively, when $X$ lies in a lower-dimensional space, we could find such a pair by PCA decomposition on $X$ to project $X$ into subspace that explains most variance of $X$. In this way, instead of considering the decomposition of $W$, we leverage the distribution of $X$ to complete the low-rank approximation. Certainly, the best way is to consider the properties of both $W$ and $X$ simultaneously, and we could mathematically present this desideratum by the following optimization problem:

$$\min_{M} \|WX - WMX\|_F^2, \quad \text{s.t.} \quad \text{rank}(M) = k, \tag{2}$$

where $M$ is the desired rank-$k$ transformation which could maximally preserve the results of matrix multiplication in the computation. In the theorem below, we will show that there exists a closed-form, optimal solution for the above optimization problem. Before stating the theorem, we first introduce some notations. Assume $\text{rank}(W) = r$ and $\text{rank}(X) = t$, we can write $W = U_W S_W V_W^T$ and $X^T = U_X S_X V_X^T$ such that

$$U_W = \begin{bmatrix} U_{W,r} & \bar{U}_{W,r} \end{bmatrix}, S_W = \begin{bmatrix} S_{W,r} & 0 \\ 0 & 0 \end{bmatrix}, V_W = \begin{bmatrix} V_{W,r} & \bar{V}_{W,r} \end{bmatrix}$$

$$U_X = \begin{bmatrix} U_{X,t} & \bar{U}_{X,t} \end{bmatrix}, S_X = \begin{bmatrix} S_{X,t} & 0 \\ 0 & 0 \end{bmatrix}, V_X = \begin{bmatrix} V_{X,t} & \bar{V}_{X,t} \end{bmatrix}.$$

In other words, the decomposition $U_W S_W V_W^T$ and $U_X S_X V_X^T$ are the full-SVD decomposition of $W$ and $X^T$. $U_{W,r}, V_{W,r}, U_{X,t}, V_{X,t}$ denote corresponding row spaces and column spaces. $\bar{U}_{W,r}$, $\bar{V}_{W,r}$, $\bar{U}_{X,t}$ and $\bar{V}_{X,t}$ are null spaces. With these notations, we are ready to state the theorem.

**Theorem 1.** Assume $\text{rank}(W) = r$ and $\text{rank}(X) = t$. The closed form solution $M^*$ of the optimization problem in equation 2 is

$$M^* = V_{W,r} S_{W,r}^{-1} Z_k S_{X,t}^{-1} V_{X,t}^T, \tag{3}$$

where $Z_k$ is the rank-$k$ truncated SVD of $Z = S_{W,r} V_{W,r}^T V_{X,t} S_{X,t}$.

The proof of Theorem 1 is postponed to the Supplementary A. We want to note that since $Z_k$ is the rank-$k$ truncated SVD of $Z$, we could also write $Z_k$ as $U_{Z,k} V_{Z,k}^T$ by distributing top-$k$ singular values of $Z$ into left or right singular matrices. Thus the original computation could be rewrote as:

$$WX \approx (WV_{W,r} S_{W,r}^{-1} U_{Z,k})(V_{Z,k}^T S_{X,t}^{-1} V_{X,t}^T) X = U^* V^* X, \tag{4}$$

where we $U^* = WV_{W,r} S_{W,r}^{-1} U_{Z,k}$ and $V^* = V_{Z,k}^T S_{X,t}^{-1} V_{X,t}^T$ are two rank-$k$ matrices, and we will replace $W$ by $U^* V^*$.

### 3.2 EXTENSION TO DOT-PRODUCT ATTENTION

Although the optimization problem in equation 2 is proposed for feed-forward computation, in this section we show that it can also be applied to dot-product part of the attention module too. The most important computation in Attention layer is to compute pairwise similarity between query and key of the sequence, and this could be described as:

$$O = (Q\bar{Y})^T (KY), \tag{5}$$

where $\bar{Y} \in \mathbb{R}^{d_1 \times n}$ is the batch query data, $Q \in \mathbb{R}^{d_2 \times d_1}$ is the query transformation matrix, $Y \in \mathbb{R}^{d_1 \times m}$ is the batch key data, $K \in \mathbb{R}^{d_2 \times d_1}$ is the key transformation matrix and $n, m$ are query and key batch size. We could again see that the desired low-rank approximation is the solution of following optimization problem:

$$\min_{M} \|(Q\bar{Y})^T (KY) - (Q\bar{Y})^T M (KY)\|_F^2, \quad \text{s.t. } \text{rank}(M) = k. \tag{6}$$

With $Q\bar{Y} = W$ and $K\bar{Y} = X$, we get the following corollary from Theorem 1 directly.

**Corollary 2.** Assume $\text{rank}(Q\bar{Y}) = r$ and $\text{rank}(KY) = t$. Denote $Q\bar{Y} = U_W S_W V_W^T$ and $(KY)^T = U_X S_X V_X^T$ the SVD decomposition of $Q\bar{Y}$ and $(KY)^T$ respectively. The closed form solution $M^*$ of the optimization problem in equation 6 is $M^* = V_{W,r} S_{W,r}^{-1} Z_k S_{X,r}^{-1} V_{X,r}^T$, where $Z_k$ is the rank-$k$ truncated SVD of $Z = S_{W,r} V_{W,r}^T V_{X,t} S_{X,t}$ .

---

**Algorithm 1:** Data-Aware Low-rank Compression of Feed-forward layer.

---

**Input:** rank $k$; training data $D_{train}$; Original weight matrix $W$; Prediction Model $M$
**Output:** Low-rank Approximation $U^*, V^*$

1  X = { }
2  **for** $x = 1, \cdots, d$ *in* $D_{train}$ **do**
3      Feed the training data $x$ into $M$ and extract the representation $\phi(x)$. $\phi(x)$ is the
        representation which will be multiplied with $W$.
4      Append $\phi(x)$ to X.
5  Given $X$,$k$ and $W$, solve the optimal low-rank matrices $U^*$,$V^*$ by equation 4.

---

### 3.3 OVERALL ALGORITHM

We have shown that the proposed DRONE method is a generic acceleration module applicable to all parts of neural language models. We summarize the DRONE on feed-forward layer in Algorithm 1. Since in practice we don't have the exact distribution of $X$, we would use training dataset to calculate the low-rank approximations as described in Algorithm 1. Attention map could be calculated by the same procedure with $W = (Q\bar{Y})^T$ and $X = KY$ as introduced above. To accelerate the whole model, we need to select appropriate ranks for each components. However, since the approximation of one component will affect the distribution of overall representations, the optimal rank for the model requires a complete search of all possible combinations of rank values, which is infeasible in practice. We thus resort to an intuitive simplification as shown in Algorithm 2 listed in the supplementary B. In short, as the changes of lower layer parameters will cause the distribution of representation shifts in upper layer, we will approximate each component in ordered sequence. In other word, we will approximate the model from the lower layers toward higher layers. Within each layer, we follow the computational sequence of underlying modules. There is a total budget parameter $r$ as an input to the Algorithm 2. The total allowed budget $r$ depends on the efficiency and efficacy trade-off which users are willing to pay. We will distribute $r$ into each module $R_{l,i}$ (allowed loss increase ratio of $i$-th module of $l$-th layer in Algorithm 2). For each module, if the approximation with certain rank used won't increase the loss over the ratio $(1 + R_{l,i})$, we will use that rank to approximate the module and move on to the next module. The distribution from $r$ to each $R_{l,i}$ is based on empirical inference time of each module. The longer a module takes to compute, the more budget would be allocated such that total allowed loss increase $(1 + r) = \prod_l \prod_i (1 + R_{l,i})$. A sample of pseudo code is provided in the supplementary to illustrate the process.

## 4 EXPERIMENTS

### 4.1 EXPERIMENTAL SETUP

We evaluate DRONE on both LSTM model and transformer-based BERT model. For LSTM, we train a 2-layer LSTM-based language model on PTB from scratch with hidden sizes $1,500$ on Penn Treebank Bank (PTB) dataset. For BERT models, we evaluate the pre-trained BERT models on GLUE tasks. Various pre-trained models are offered in the open source platform (Wolf et al., 2019). For BERT models, we use BERT-base models and it contains 12 layers of the same model structure without sharing parameters. Each layer contains an attention module with hidden size 768 and 12 channels, a small $768 \times 768$ Feed-forward (FF) layer followed by 2 larger FF layers ($768 \times 3,072$ and $3,072 \times 768$). As shown in Figure 1, these four components contribute to the most computational time in the BERT-base models.

To the best of our knowledge, we are the first work to discuss the generic matrix computation acceleration on NLP tasks. Therefore, our baseline comparison will be the SVD low-rank approximation. We will also include those state-of-the-art distillation methods TinyBERT (Jiao et al., 2019) in the comparison and show that the proposed method could be combined with it to further improve the performance. TinyBERT reduces the model into 4 layers of attention dimension 312 with 12 channels, and the FF layers are downsized to $312 \times 1,200$.

As we mentioned above that all the approximation methods need to consider efficiency and efficacy trade-off. In principle, we would have a plot of accuracy versus speedup ratio as shown in Figure 3 for MRPC and SST-2 tasks. The allowed accuracy loss is up to tolerance of individual users. In this paper, we will report the approximation results with about $3\%$ loss of accuracy. The inference

| Methods | MNLI | QQP | SST-2 | QNLI | MRPC | RTE | CoLA | STS-B |
|---|---|---|---|---|---|---|---|---|
| Original | 84.3 | 90.9 | 92.3 | 91.4 | 89.5 | 72.6 | 53.4 | 87.8 |
| SVD | 74.4 | 50.8 | 73.1 | 52.2 | 63.8 | 47.3 | 26.0 | 0.13 |
| DRONE | 82.0 | 89.4 | 90.0 | 88.5 | 86.7 | 70.0 | 52.5 | 85.8 |
| DRONE-Retrain | 82.6 | 90.1 | 90.8 | 89.3 | 88.0 | 71.5 | 53.2 | 87.8 |
| Speedup Ratio | 1.60x | 1.25x | 1.64x | 1.20x | 1.92x | 1.31x | 1.33x | 1.52x |

Table 1: The experimental results of natural language inference tasks on Glue dataset.

| Tasks | Self-Attention | Feed-Forward 0 | Feed-Forward 1 | Feed-Forward 2 | Others | Total Time |
|---|---|---|---|---|---|---|
| MNLI | 122.7 | 19.5 | 78.5 | 46.1 | 4.2 | 268.0 |
| QQP | 131.5 | 29.9 | 99.2 | 66.5 | 5.8 | 333.0 |
| SST-2 | 100.5 | 24.7 | 79.3 | 54.5 | 4.5 | 263.6 |
| QNLI | 128.3 | 28.4 | 111.0 | 79.0 | 5.9 | 352.6 |
| MRPC | 82.6 | 12.8 | 89.4 | 38.2 | 2.4 | 225.3 |
| RTE | 116.0 | 25.6 | 85.4 | 62.3 | 3.4 | 292.7 |
| CoLA | 108.2 | 22.7 | 93.1 | 70.8 | 3.4 | 298.3 |
| STS-B | 109.1 | 19.3 | 90.8 | 53.0 | 4.0 | 276.2 |

Table 2: The detailed average inference time of each component in the model by retrained DRONE method. The unit is in millisecond.

| Tasks | Models | Self-Attention | Feed-Forward 0 | Feed-Forward 1 | Feed-Forward 2 | Others | Total Time | speedup | Accuracy (%) |
|---|---|---|---|---|---|---|---|---|---|
| STS-B | BERT | 139.7 | 32.5 | 134.7 | 109.0 | 4.4 | 420.2 | 1x | 87.8 |
| | TinyBERT | 18.2 | 2.0 | 9.7 | 6.2 | 0.6 | 36.7 | 11.4x | 86.9 |
| | DRONE | 16.1 | 1.7 | 7.5 | 3.4 | 0.8 | **29.5** | **14.2x** | **87.0** |
| RTE | BERT | 124.3 | 27.7 | 129.0 | 101.5 | 3.1 | 385.5 | 1x | 72.6 |
| | TinyBERT | 70.2 | 16.2 | 66.4 | 54.3 | 3.1 | 210.2 | 1.8x | 70.8 |
| | DRONE | 63.8 | 16.6 | 58.9 | 50.0 | 2.8 | **185.0** | **2.1x** | **71.7** |
| MRPC | BERT | 131.6 | 29.9 | 140.4 | 127.6 | 3.0 | 432.5 | 1x | 89.5 |
| | TinyBERT | 18.6 | 1.9 | 9.9 | 6.1 | 0.7 | 37.2 | 11.6x | 86.3 |
| | DRONE | 17.4 | 1.8 | 9.9 | 5.5 | 0.7 | **35.3** | **12.3x** | **86.7** |
| SST-2 | BERT | 146.3 | 36.1 | 133.6 | 110.0 | 5.3 | 431.3 | 1x | 92.3 |
| | TinyBERT | 18.1 | 1.9 | 9.9 | 6.2 | 0.7 | 36.8 | 11.7x | **90.7** |
| | DRONE | 14.4 | 1.9 | 7.8 | 3.5 | 0.5 | **28.1** | **15.3x** | **90.7** |

Table 3: The average inference time of comparison to distilled models. The unit is in millisecond.

speed is measured on an Intel(R) Xeon(R) CPU E5-2640 v4 @ 2.40GHz chip with single-thread. Inference are performed in a batch fashion with size 100 of the whole evaluation dataset. Average single sequence prediction inference time in millisecond is reported in the results. To perform the approximation, we randomly sub-sample $10\%$ of the training data to be the training distribution used. After the proposed data-aware low-rank distribution, we could slightly fine-tune the model to make the performance better. We use a relatively smaller learning rate $10^{-7}$ and retrain 1 epoch on the sub-sampled training data to complete the fine-tuning procedure.

## 4.2 RESULTS OF BERT MODELS ON GLUE DATASET

We summarize the results of the DRONE on GLUE tasks in Table 1. We could observe that each task exhibits different difficulty. The best acceleration we could achieve is $92\%$ faster with less than $2\%$ accuracy loss after retraining (MRPC). In addition, DRONE achieves $1.52x$ acceleration without accuracy loss on the STS-B task. By applying the same selected rank for each module with SVD method, we could observe that the performance drops significantly. This shows that the matrices within the model is generally not low-rank; thus the direct low-rank approximation without considering data distribution won't work. The detailed average inference time of the approximated models are listed in Table 2. We could observe that the FF2 layer could be accelerated most. A plausible reason could be that the input dimension to the FF2 layer is in a larger dimension (3072) than all the other layers (64 or 768). When the input distribution actually lies in a lower-dimensional space, there is much more room for FF2 layer to be compressed and accelerated by the data-aware low-rank method.

## 4.3 COMBINATION WITH MODEL SIZE REDUCTION METHODS

Distillation is a competitive method to compress the underlying model into a smaller one without losing much accuracy. Distilled models are much smaller in number of layers or hidden dimension, resulting a smaller model size and faster inference time. As shown in the Table 3, TinyBERT, one of the most competitive distillation methods, indeed achieves good performance within $3\%$ accuracy

| Tasks | Models | Total Time | speedup | Accuracy (%) |
|---|---|---|---|---|
| STS-B | BERT | 3.17 ms | 1x | 87.8 |
| | TinyBERT | 0.37 ms | 8.6x | 86.9 |
| | **DRONE** | **0.29 ms** | **10.9x** | **87.0** |
| RTE | BERT | 3.34 ms | 1x | 72.6 |
| | TinyBERT | 1.73 ms | 1.9x | 70.8 |
| | **DRONE** | **1.51 ms** | **2.2x** | **71.7** |
| MRPC | BERT | 3.28 ms | 1x | 89.5 |
| | TinyBERT | 0.42 ms | 7.8x | 86.3 |
| | **DRONE** | **0.38 ms** | **8.6x** | **86.7** |
| SST-2 | BERT | 3.29 ms | 1x | 92.3 |
| | TinyBERT | 0.39 ms | 8.4x | **90.7** |
| | **DRONE** | **0.34 ms** | **9.7x** | **90.7** |

Table 4: The average inference time of comparison to distilled models on GPU. The unit is in millisecond.

loss for some of the GLUE tasks. We want to emphasize that DRONE is orthogonal to distillation methods. Since DRONE is generic, it could be combined together with other methods to further improve the performance. Due to the fact that the computation inside the distilled model is still full matrix computation, DRONE could then be applied to find the data-aware low-rank approximation of these smaller matrices. Results are summarized in Table 3. As we can see that combined with the distillation method, DRONE could further reduce the inference time without sacrificing accuracy. In particular, on SST-2 task DRONE speedups the inference time from 11.7x to 15.3x while achieving the same accuracy as the TinyBERT. These results again show that the proposed method is generic and has the potential to be applied under various scenarios.

### 4.4 Results on LSTM models

We demonstrate that DRONE could also work on accelerating matrices in the LSTM model. As shown in Table 5 in the Supplementary C, DRONE could accelerate 2-layer LSTM models about 3.4x on PTB dataset. And the result is slightly better than SVD methods. After the fine-tuning, DRONE could achieve less than $1\%$ accuracy loss. This shows that DRONE is generic. As long as the underlying computation is a matrix multiplication, DRONE could leverage the data distribution to obtain a better low-rank approximated computation.

### 4.5 Could the low-rank Structure learned by End-to-end training?

One natural question to ask is if once the rank is decided, the same optimal low-rank structure could be learned by end-to-end fine-tuning. We have conducted the experiments on MRPC to verify this, and the results are summarized in Table 6 in the Supplementary D. We start the fine-tuning from the SVD results, and use the fine-tuning hyper-parameters as in (Wolf et al., 2019)[1]. After fine-tuning, accuracy goes from $63.8\%$ to $85.8\%$ which is still slightly worse than DRONE. This shows that fine-tuning on the SVD result might not achieve the best low-rank result. The proposed method under the the optimization problem (equation 2) indeed provides a good initial solution to the data-aware low-rank problems.

## 5 Conclusion

In this work, we propose DRONE, a data-aware low-rank approximation, to achieve a better low-rank approximation. DRONE leverages the fact that data distribution in NLP tasks usually lies in a lower-dimensional subspace. By including the data distribution into consideration, we propose a data-aware low-rank approximation problem and provide an closed-form solution. Empirical results validates that DRONE could significantly outperformed the vanilla-SVD method. It could achieve at least $20\%$ acceleration with less than $3\%$ accuracy loss. When combined with the distillation methods, DRONE could achieve 15.3 times acceleration with less then $2\%$ accuracy loss.

---

[1]https://huggingface.co/transformers/v2.1.1/examples.html#glue

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

# Supplementary Material for:
# Data-Aware Low-Rank Compression for Large NLP Models

## A    PROOF OF THEOREM 1

**Theorem 1.** Assume $\text{rank}(W) = r$ and $\text{rank}(X) = t$. The closed form solution $M^*$ of the optimization problem in equation 2 is

$$M^* = V_{W,r} S_{W,r}^{-1} Z_k S_{X,t}^{-1} V_{X,t}^T, \tag{7}$$

where $Z_k$ is the rank-$k$ truncated SVD of $Z = S_{W,r} V_{W,r}^T V_{X,t} S_{X,t}$.

*Proof.* We firstly consider the unconstrained problem:

$$
\begin{aligned}
M^* &= \underset{M}{\arg\min} \, \|WX - WMX\|_F^2 \\
&= \underset{M}{\arg\min} \, \|U_W^T W X U_X - U_W^T W M U_X\|_F^2 \\
&= \underset{M}{\arg\min} \, \|S_W V_W^T V_X S_X - S_W V_W^T M V_X S_X\|_F^2,
\end{aligned}
$$

where the second equality holds due to the fact that $U_W$ and $U_X$ are orthonormal matrices. Note that we could expand the term $S_W V_W^T V_X S_X$ as:

$$
\begin{aligned}
S_W V_W^T V_X S_X &= \begin{bmatrix} S_{W,r} & 0 \\ 0 & 0 \end{bmatrix} \begin{bmatrix} V_{W,r}^T \\ \bar{V}_{W,r}^T \end{bmatrix} \begin{bmatrix} V_{X,t} & \bar{V}_{X,t} \end{bmatrix} \begin{bmatrix} S_{X,t} & 0 \\ 0 & 0 \end{bmatrix} \\
&= \begin{bmatrix} S_{W,r} V_{W,r}^T & 0 \\ 0 & 0 \end{bmatrix} \begin{bmatrix} V_{X,t} S_{X,t} & 0 \\ 0 & 0 \end{bmatrix} \\
&= \begin{bmatrix} S_{W,r} V_{W,r}^T V_{X,t} S_{X,t} & 0 \\ 0 & 0 \end{bmatrix}.
\end{aligned}
$$

Similarly, we will have

$$
S_W V_W^T M V_X S_X = \begin{bmatrix} S_{W,r} V_{W,r}^T M V_{X,t} S_{X,t} & 0 \\ 0 & 0 \end{bmatrix}.
$$

Therefore, we could continue above unconstrained problem as:

$$
\begin{aligned}
M^* &= \underset{M}{\arg\min} \, \|S_W V_W^T V_X S_X - S_W V_W^T M V_X S_X\|_F^2 \\
&= \underset{M}{\arg\min} \, \left\| \begin{bmatrix} S_{W,r} V_{W,r}^T V_{X,t} S_{X,t} - S_{W,r} V_{W,r}^T M V_{X,t} S_{X,t} & 0 \\ 0 & 0 \end{bmatrix} \right\|_F^2 \\
&= \underset{M}{\arg\min} \, \|S_{W,r} V_{W,r}^T V_{X,t} S_{X,t} - S_{W,r} V_{W,r}^T M V_{X,t} S_{X,t}\|_F^2. \\
&= \underset{M}{\arg\min} \, \|Z - S_{W,r} V_{W,r}^T M V_{X,t} S_{X,t}\|_F^2.
\end{aligned}
$$

The above minimization problem obtains the optimal value if $S_{W,r} V_{W,r}^T M V_{X,t} S_{X,t}$ equals the rank-$k$ truncated SVD of Z by the fundamental property of SVD decomposition. Thus, we will have:

$$
Z_k = S_{W,r} V_{W,r}^T M^* V_{X,t} S_{X,t} \implies M^* = V_{W,r} S_{W,r}^{-1} Z_k S_{X,t}^{-1} V_{X,t}^T.
$$

$\square$

## B  AN ALGORITHM TO SEARCH OF RANKS UNDER DRONE

In Algorithm 2, we illustrate how to select the rank of each module by applying DRONE illustrated in Algorithm 1. The input to Algorithm 2 consists of training data, the model with all parameters of weight matrices and original training loss. In addition, a pre-defined search grid is also necessary. Taking $W \in R^{768 \times 768}$ as an example, we can perform a grid search for a proper low rank $k$ over $[1, 768]$ such as $\{96, 192, 288, 384, \ldots, 768\}$. The finer the grid, the more compressed model we could get at the cost of longer running time of the DRONE method. With these input parameters, we firstly distribute the total allowed loss into each individual module. We then iteratively apply Algorithm 1 following the computational sequence illustrated in Figure 1. For the compression of each module, we search the rank $k$ by going through the grid. If the approximated result will not increase the allowed loss increase ratio of the component, we will end the search and tie the found rank to the component and move on. The procedure will continue until all components are compressed. The whole process could guarantee us that the final loss $L'$ of the compressed model $\hat{M}$ would not be greater than $(1 + r)L$, where $L$ is the original loss before approximation.

---

**Algorithm 2:** Overall Algorithm of Grid Search of Low-rank Model Approximation

---

**Input:**  training data $D_{train}$; Original weight matrix $W$; Prediction Model $M$, total allowed loss increase ratio $r$, Search grids of ranks for each module $G$, original Training loss $L$,

**Output:** Low-rank Model $\hat{M}$

1   $R \leftarrow$ Distribute allowed ratio $r$ into each module.
2   **for** $l = 1, \cdots, $ *total layers* **do**
3      **foreach** *module* $m_i \in M_l$ **do**
4         $W_{l,i} \leftarrow$ $l$-th layer parameter of module $m_i$ (e.g., 2nd feed-forward matrix in first layer.)
5         **for** $i = 1, \cdots, |G_{l,i}|$ **do**
6            $k \leftarrow G_{l,i}$
7            $U, V \leftarrow$ Algorithm 1 $(k, D_{train}, W_{l,i}, M)$
8            $\hat{M} \leftarrow M$ with $W_{l,i}$ replaced by $U, V$.
9            Evaluate new loss $L_{new} = \hat{M}(D_{train})$
10           **if** $L_{new}/L < 1 + R_{l,i}$ **then**
11              $M \leftarrow \hat{M}$
12              break

---

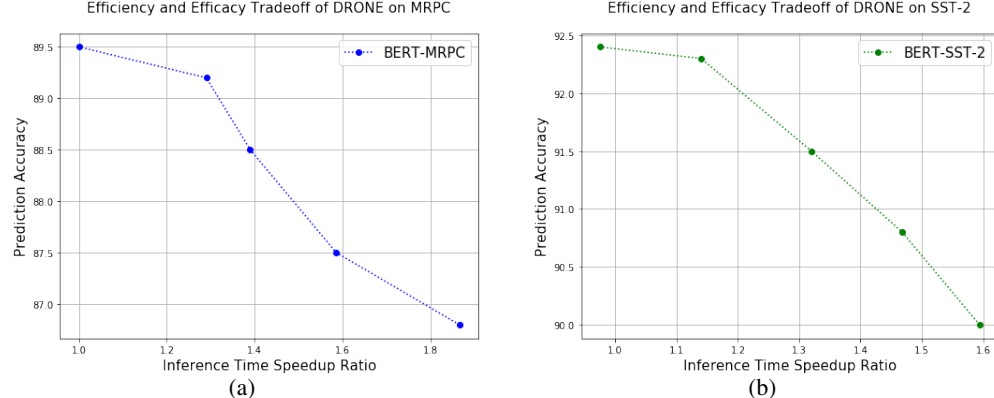

Figure 3: Illustration of efficiency and efficacy trade-off. Each point in this graph represents a specific ratio of training loss increase after approximation.

## C  EFFICIENCY AND EFFICACY TRADE-OFF GRAPH

## D  LSTM RESULT

| Models | LSTM-1 | LSTM-2 | Softmax | Others | Total Time | Perplexity |
|---|---|---|---|---|---|---|
| PTB-Large | 1.27ms | 1.30ms | 1.09ms | 0.13ms | 3.79ms | 78.32 |
| PTB-Large-SVD | - | - | - | - | - | 81.09 |
| PTB-Large-DRONE | - | - | - | - | - | 80.87 |
| PTB-Large-DRONE-Retrain | 0.24ms | 0.34ms | 0.42ms | 0.11ms | 1.11ms(3.4x) | 79.01 |

Table 5: The average inference time of each component in the model of 2-layer LSTM model. Both proposed methods and SVD use same ranks so the inference time is approximately the same. The unit is in millisecond and the number in parenthesis shows the ratio respective to the overall inference time.

## E  RESULTS OF SVD-BASED RETRAINING

| Models | Accuracy (%) |
|---|---|
| BERT-MRPC | 89.5 |
| BERT-MRPC-DRONE | 86.8 |
| BERT-MRPC-SVD | 63.8 |
| BERT-MRPC-SVD-Retrain | 85.8 |

Table 6: Illustration of SVD fine-tuning. Using the same rank as the proposed method, SVD accuracy will drop significantly. After fine-tuning on the SVD-based approximation, the accuracy could be recovered. But it's still less competitive than the proposed method.

## F  PYTHON PSEUDO CODE OF SOLVING EQUATION 2

```
import numpy as np

def OPTsolver(x,y,k):
    '''
    compute the best rank k projection M such that \| x*y' - x*M*y'\|_{F
    } is minimized
    x \in shape n x d
    y \in shape m x d
```

```
 9      '''
10      xSS = np.matmul(x.transpose(),x)
11      kSS = np.matmul(y.transpose(),y)
12      U1,S1,V1 = np.linalg.svd(xSS,False)
13      S1 = S1 ** 0.5
14      I1 = np.eye(S1.shape[0])
15      U2,S2,V2 = np.linalg.svd(kSS,False)
16      S2 = S2 **0.5
17      I2 = np.eye(S2.shape[0])
18      YK = np.dot(np.dot(I1*S1,V1),np.dot(V2.transpose(),I2*S2))
19      U,S,V = np.linalg.svd(YK,False)
20      L = np.dot(V1.transpose(),I1*(1/S1))
21      R = np.dot(I2*(1/S2),V2 )
22      M = np.dot(U[:,:k]*S[:k],V[:k,:])
23      return L,R,U,S,V
```

Listing 1: The python function to solve the equation 2.

## G PYTHON PSEUDO CODE OF RANK SEARCHING

```
 1
 2 import os
 3 import numpy as np
 4 import torch
 5 import subprocess as sp
 6
 7 cuda_num = 7
 8 n_heads = 12
 9 total_layer = 12
10
11 prev_loss = .11159391902588509 # Initial Loss
12 the_model_name = 'bertSST2'
13
14 time_attn = 117.5 # Empirical Inference Time on Attention Module
15 time_0 = 34.27 # Empirical Inference Time on Attention FFL Module
16 time_1 = 133.11 # Empirical Inference Time on Feedforward 1 layer
17 time_2 = 128.84 # Empirical Inference Time on Feedforward 2 layer
18 minimal_time = min(time_attn,time_0,time_1,time_2)
19 multiplier = (time_attn+time_0+time_1+time_2)/(minimal_time)
20 tolerant = 2. # allowed loss increase ratio. $r$ in Algorithm 2.
21
22 # Code to Distribute the $r$ into individual Modules.
23 # The distribution depends on empirical inference time of each module and
        number of layers.
24 basic_tolerance = np.exp(np.log(tolerant)/multiplier)
25 tol_attn = np.exp(np.log(basic_tolerance**(time_attn/minimal_time))/
       n_layer)
26 tol_0 = np.exp(np.log(basic_tolerance**(time_0/minimal_time))/n_layer)
27 tol_1 = np.exp(np.log(basic_tolerance**(time_1/minimal_time))/n_layer)
28 tol_2 = np.exp(np.log(basic_tolerance**(time_2/minimal_time))/n_layer)
29
30
31 #### Omitted Code ###
32 # This part of the code is to change some parameters of the underlying
       hugginface framework in order to extract the training distribution X
       of each module from the model.
33 ### Omitted Code ###
34
35
36
37 for i in range(total_layer):
38     for each module in the layer: # This line is pseudo code for clarity
       reason.
39         # This part of the code extracts $R_{l,i}$(named the_tol here) in
        Algorithm 2.
40         if save_symbol == "E":
```

```
41              the_tol = tol_attn
42          elif save_symbol == "F0":
43              the_tol = tol_0
44          elif save_symbol == "F1":
45              the_tol = tol_1
46          else:
47              the_tol = tol_2
48          # Update the allowed increase of loss
49           prev_loss = prev_loss * the_tol
50
51          rank = 16 if save_symbol == "E" else 96 # initial search rank for
       Attention(16) and FFL layers(96)
52          tps = 64 if save_symbol == "E" else 768 # Maximal rank specified
      in the original models.
53          while rank <= tps:
54              ### Omitted Code ###
55              ## Write the tried rank into hugginface framework##
56              ### Omitted Code ###
57
58              # This line run the inference in the command line
59              os.system('CUDA_VISIBLE_DEVICES="'+str(cuda_num)+'" python
      run_glue.py --model_type bert --model_name_or_path /data/TinySeries/
      SST2/OriginalSST2/ --task_name SST-2 --do_eval --data_dir /data/
      glue_data/SST-2/ --output_dir /tmp/sst-2 --per_gpu_eval_batch_size
      100 --per_gpu_train_batch_size 100 --max_seq_length 128 > /tmp/tmp0')
60
61
62              with open('/tmp/tmp0','r') as file:
63                  data = file.readlines()
64              new_r = float(data[-1])
65              if  new_r < prev_loss:
66                  break
67              if save_symbol == "E": # Attention module, we increase search
       rank 16 at a time.
68                  rank += 16
69              else:
70                  #rank += 96 # For FFL layer, we increase search rank 96 a
      time.
71                  if rank ==  384:
72                      rank = 768
73                      break
74                  else:
75                      rank += 96
76
77          ### Omitted Code ###
78          # This part of code update the model #
```

Listing 2: A mixed of real code and pseudo code to illustrate the search algorithm.

