# OpenReview forum: "Data-aware Low-Rank Compression for Large NLP Models"
_ICLR.cc/2021/Conference — Reject_

### Official Review · AnonReviewer2 · 2020-10-28
**Official Blind Review #2**

**Rating:** 6
**Confidence:** 4

**Review:**

This work introduces a low-rank based compression method, called DRONE, to accelerate the inference of large NLP models. Instead of decomposing weight matrices in a model, DRONE proposes to exploit the low-rank decomposition by considering the input vectors, which can be in a low-rank space. To compress the whole model, DRONE performs low-rank decomposition from the lower layers to the upper layers in a sequential order. DRONE also proposes a way to extend the low-rank decomposition to dot-product attention. In the evaluation, the paper evaluates their approach against BERT-base and LSTM, and shows that their approaches can obtain better latency-vs-accuracy trade-off versus SVD-based compression method.

Strengths:
- The paper aims to address an important problem in large NLP model inference.
- It is an interesting idea to apply matrix decomposition by exploiting the low-rank in the input data rather than the weight matrix.

Weakness:
- The evaluation is inadequate as the baseline does not seem to represent real inference scenarios.
- Important references are missing, making it not clear the advantage of this work as compared with existing approaches that also compress Transformer networks.

Comments:

The idea of doing data-aware low-rank compression is certainly interesting. However, the evaluation is largely inadequate to make it convincing that the data-aware low-rank compression will bring a benefit in NLP inference.

First, it is not clear why a large batch size (e.g., 100) is chosen for inference. In real online inference scenarios, queries come in one-by-one, so the batch size is often just 1 or at most a few in order to meet latency constraint. Is large batch size chosen because DRONE cannot obtain much speedup with small batch sizes? What is the latency when batch size equals to 1? It would be better to add discussions on the choice of batch sizes and show speedup ratios using batch sizes that are close to real settings.

Second, the evaluation is done "with single-thread on CPU", which does not represent real scenario either and makes the whole evalution less convincing. Modern servers often have multi-core CPUs and GPU with thousands of cores, and DL frameworks often support parallel execution on different hardwares. One caveat is that reducing the width of matrix-multiplication, e.g., using low-rank decomposition, may not necessarily lead to significant latency reduction, as general matrix-multiplication can execute quite efficiently through highly optimized GEMM kernels (e.g., in cuBLAS and MKL) on GPU and CPU. Therefore, it is important to report the performance of DRONE with parallelism enabled. It is important because if the latency reduction is relatively small on real hardware setting, it becomes difficult to justify the 3% accuracy loss on downstream tasks using compression.

Finally, it does not seem to be accurate to claim "the first work to discuss the generic matrix computation acceleration on NLP tasks", given that low-rank decomposition has been used quite a bit for model compression (e.g., SVD and its variants), and there is at least some work on applying low-rank decomposition for compressing Transformer networks[1] and LSTM[2,3].

Question:

The empirical observation that weight matrices in BERT models are not low-rank is interesting. Do you have an explanation on why this is the case?

Does DRONE require to pre-decide a fixed batch size to apply the closed-form solution as in Equation 3? How does the speedup ratio look like when the batch size is 1?

Minor:
[3] applies low rank decomposition for LSTMs, not CNN.

[1] Mao et. al. "LadaBERT: Lightweight Adaptation of BERT through Hybrid Model Compression", https://arxiv.org/pdf/2004.04124.pdf, April 2020

[2] Winata et. al. "On the Effectiveness of Low-Rank Matrix Factorization for LSTM Model Compression",  https://arxiv.org/abs/1908.09982, 2019

[3] Yang et. al. "Fast LSTM Inference by Dynamic Decomposition on Cloud Systems", https://ieeexplore.ieee.org/document/8970702/, ICDM, 2019

---

> ### Author Response · Authors · 2020-11-22
> **Experimental setup is further clarified and GPU results are added. The speedup is still prominent.**
>
> 1. We want to clarify that despite using a batch of size 100, we limit the thread to be 1 thus the inference time of batch size 1 is roughly the same as the ones listed in the tables/figures. When the thread is not limited to 1, it’s indeed the case as reviewer pointed out that BLAS and MKL would come into work and the overall time for all methods will be improved. Specifically, all methods are about 8 times faster. Take STS-B for example, the original BERT improved from 420 ms to 50 ms and the proposed method improved from 29.5 ms to 4.1 ms. There is still over 10x faster on multi-thread environment.  We use this setup as this is a common scenario when the inference is done on devices (e.g., mobile phone or IoT edge devices). In these scenarios, number of cores is rather limited and multi-thread could still be costly. Since DRONE method itself is a dimension reduction scheme, a smaller model will certainly lead to a better acceleration.  We chose an evaluation setup such that the speedup ratio could genuinely reflect the change of model sizes. The GPU results are updated in the revised Table 4. Basically it still could achieve a 10x acceleration on various GLUE tasks. When the case is GPU, batch size = 1 could be harmful as GPU’s power would be exploited when the same instruction is operated on multiple instance of data. We need to collect at least 10 points to achieve the reported results on GPU. When the inference is on device, it’s usually the case as reviewer pointed out that the inference is done one-by-one. But when it’s done on GPU server in industry facilities, usually the service would collect multiple queries together to make the batch size larger in order to better leverage GPU. In sum, the acceleration speedup ratio is hardware dependent but overall the speedup trend is the same as DRONE could indeed reduce the model size to achieve the acceleration.
>
>
>
> 2. We thank reviewer for pointing out there are other low-rank applications works. We will add it to the related works.
>
> 3. On the possible explanation of non low-rank. We believe it’s a problem pertaining to all model/task/data. We think there might be cases such that low-rank structure exists in a specific model/task/data combination, but it’s unlikely an universal phenomena.  Essentially a low-rank structure exists means that the specified dimensions are too large to represent the information in each token. As training data size increases in training transformer, it’s more likely that the model dimension needs to be increased rather than decreased. This prohibits a substantial possibility of low-rank. Since we hypothesize that each dimension carries certain types of information, it’s likely that after some clustering we could have low-rank structure in each cluster but not collectively low-rank.

---

> > ### Comment · AnonReviewer2 · 2020-11-23
> > **Comments**
> >
> > Thank you for the detailed response. However, there are a few arguments in the authors' response that I do not believe to be true, although I'm happy to be convinced otherwise.
> >
> > Regarding the batch size, the authors say, "despite using a batch of size 100, we limit the thread to be 1 thus the inference time of batch size 1 is roughly the same as the ones listed in the tables/figures." This "roughly the same" argument seems problematic. In fact, they are often quite different because taking a batch size of 100 allows increasing the amount of data reuse during computation significantly. To execute a batch of input samples, at any given time, one only needs to load a subset of the model weights (e.g., one layer) into the fast memory (e.g., cache) to perform the computation. In contrast, if the batch size is 1, such data loading overhead might be triggered for every input sample. Since the data movement cost can be orders of magnitude larger than the compute cost, especially when the authors further limit the compute capacity to just 1 core, does running a batch size of 100 with 1 thread really give us equivalent inference time as running a batch size of 1 with 1 thread? The authors need to provide results to justify this argument.
> >
> > Second, regarding the target devices, there was nowhere in the original paper that emphasized that the technique targeted mobile phone or IoT edge devices. Yet, the response says, "We use this setup as this is a common scenario when the inference is done on devices (e.g., mobile phone or IoT edge devices). In these scenarios, the number of cores is rather limited and multi-thread could still be costly." However, the hardware used for evaluation was Intel(R) Xeon(R) CPU E5-2640 v4, which is a multi-core CPU. Therefore, the paper seems to perform evaluation by explicitly limiting the number of cores, which seems to be unrealistic. Later, the authors added results on GPU, which starts to confuse me as whether the proposed technique is for mobile/IoT devices or for accelerators with parallel comput units.  The authors may need to better position when the proposed technique is helpful.
> >
> > Finally, there are several descriptions in the rebuttal that are unclear. For example:
> > - "Specifically, all methods are about 8 times faster.", but "faster" than what?
> > - "We need to collect at least 10 points to achieve the reported results on GPU." What does "points" mean here? Do you mean a batch size of 10?
> > - "Take STS-B for example, the original BERT improved from 420 ms to 50 ms and the proposed method improved from 29.5 ms to 4.1 ms. There is still over 10x faster on multi-thread environment. ", I don't do not see where this "over 10x faster" comes from.
> > - "But when it's done on GPU server in industry facilities, usually the service would collect multiple queries together to make the batch size larger in order to better leverage GPU. " This is true but under a condition, which is if each of the batched queries can meet a latency requirement for inference. If, for individual queries (batch size 1), the proposed technique is slower than the original model, and the benefits only start to show when the batch size is larger than a certain threshold (e.g., >10), then the authors may need to justify why how its technique can be integrated with an existing inference pipeline.

---

> > > ### Author Response · Authors · 2020-11-24
> > > **Million thanks for the reviewer continuing this constructive discussion.   Details are explained.**
> > >
> > >
> > > Firstly we'd like to express the Million thanks for the reviewer continuing this constructive discussion. We will be answering accordingly to the bullets listed in the response.
> > >
> > >
> > > 1) We understand what the reviewer means and would like to summarize it again our understanding.
> > >
> > > Basically all models and data are loaded into RAM. To execute it, we need to move model/data into the cache and ALU in order to complete the operation. Nowadays, we usually use some framework such as tf or pytorch and these operations are handled by the framework. Unless digging into the framework deeply, this sort of data/model move would be handled by the framework and we don’t directly control it. Thus, we cannot directly know how long it takes to move the data/model. And the reviewer has the concern that moving between RAM and Cache/ALU could be the bottleneck of the whole computation process. In this situation the best thing we could do is to provide the empirical result to validate the time cost of data/model movement.
> > >
> > > Empirically we evaluate this (original non approximated model) on STS-B and SST-2 dataset. STS-B with 1-thread/batch-100 has on average inference time 420.2ms and 1-thread/batch-1 is 409.5 ms. SST-2 with 1-thread/batch-100 on average inference time is 431.3ms,  SST-2 with 1-thread/batch-1 420.45356456292876. It’s true that different batch size of data could affect the inference time a bit. But it’s roughly in the same order. For distilbert and DRONE the difference between batch-1 and batch-100 is smaller than 1 ms. We will be adding an additional table in the Appendix but based on the results on 2 dataset number of threads won’t be a major concern empirically when we compare the speedup ratio, and either one could be reflecting the trend of speedup achieved by DRONE.
> > >
> > >
> > > 2. Sorry for the confusion but we want to emphasize again that the proposed method is generic to all the devices. We have added GPU results for the completeness but we do not emphasize the method is dedicated for IoT/mobile/CPU/GPU. In fact, DRONE is applicable to all the situations as it in essence is a better low-rank approximation technique to a much smaller model size. This is the most important argument and the more lightweight model will lead to faster inference on all sorts of hardware.
> > >
> > >
> > >
> > > Remaining clarification of the rebuttal statements.
> > >
> > > -   On 8x faster.
> > >
> > > Using multiple threads on CPU would enhance all methods as compared to evaluation on CPU with 1 thread. Thus, 8x faster means all methods performed by multiple-threads are roughly 8 times faster than evaluating with 1-thread. Each method only compares to itself and the difference is the number of threads used.
> > >
> > > - On the usage of word "points."
> > >
> > > Yes we mean the batch size of 10. Sorry for the confusion.
> > >
> > > - On clarifying question. "Take STS-B for example, the original BERT improved from 420 ms to 50 ms and the proposed method improved from 29.5 ms to 4.1 ms. There is still over 10x faster on multi-thread environment. "
> > >
> > > In the paper, we evaluated the result by using only 1 thread. For STS-B with  original unapproximated transformer it will achieve on average inference time 420.2ms for   1-thread/batch-100  and 409.5 ms for 1-thread/batch-1. If we allow multiple threads, batch-1 will have 103ms and batch-100 on average is 50.1ms. For the DRONE model, without multiple thread it’s 29.5 ms and with multiple threads it will have 4.1ms inference time. So initially for single thread situation we have a 420.2/29.5 ~= 14.2x faster inference time (for single thread environment). And if we consider the speedup in multi-threads environment, if a 50.1/4.1 ~= 12.2x speedup, which is still a speedup rate larger than 10x. That means even in the multi-thread environment, DRONE could achieve the same order of speedup as in single-thread environment.
> > >
> > > - On GPU with batch size 1
> > > Firstly we want to clarify that it will not be slower than an unapproximated model. Take STS-B for example, DRONE alone will have 14.6ms inference time and original BERT is 15.3ms. When combining with distilbert, DRONE will achieve 2.24ms (distilbert alone is 2.3ms). So the worst case scenario the DRONE will degrade to the unapproximated model and non prominent speedup observed. A possible fix is actually digging into hardware management. Packages such as TensorRT from nVidia could play a role in this.

---

> > > > ### Comment · AnonReviewer2 · 2020-11-24
> > > > **Comments**
> > > >
> > > > Thank you for providing a detailed response. Adding batch size 1 and multiple threading results is helpful. The >10X speedups seem to be more of an effect of the combination of the distillation and low-rank approximation.  Although the accuracy seems to be slightly higher than TinyBERT, the actual speedups seem to be quite marginal compared with TinyBERT.  Also,
> > > > it is still not clear to me whether DRONE requires to pre-decide the batch size to apply the closed-form solution as in Equation 3, which would have an impact on its practical use.
> > > >
> > > >  I increased my score, as I appreciate the novel idea of the data-aware low rank compression, the authors' effort to show that it works collaboratively with distillation, and its added evaluation results on multi-threading CPU and GPUs. I also thank the authors for their detailed responses to address my questions.

---

### Official Review · AnonReviewer3 · 2020-10-29

**Rating:** 5
**Confidence:** 4

**Review:**

Summary:
This paper studies a technique to increase the inference speed and decrease model sizes of pretrained NLP models such as BERT. Since most operations in BERT consist of matrix multiplications, the authors conduct empirical experiments to show that while matrices themselves are not low-rank, the learned representation of each layer lies in a low-dimensional space. These empirical insights lead the authors to propose an approach based on data-aware low-rank compression of pretrained weight matrices and can be applied to fully-connected and self-attention layers. Their approach is able to improve both model size and inference speed
with limited loss of accuracy on the Glue dataset

Strengths:
1. The paper studies an important problem and is well-motivated. The writing is generally clear and supported by figures and experimental results.
2. The algorithm is presented very clearly and is very intuitive. Their algorithm is also supported by examples and proofs when necessary.
3. The proposed approach is simple to implement and can be applied for different network architectures. This is good and the results are also promising, but there are some issues with evaluation (see below).
4. Pseudocode is provided in the appendix and the approach seems easy to implement and quite flexible.

Weaknesses:
1. Figure 3 tradeoff between performance and speedup is not very useful unless the plots are also made for existing baselines and compared with the author's proposed approach.
2. The only empirical comparison is with TinyBERT but it seems like there are many recent papers studying similar compression algorithms on BERT:
https://arxiv.org/abs/1909.11687
https://arxiv.org/abs/2001.04246
https://arxiv.org/abs/2002.02925
https://arxiv.org/abs/2002.08307
that I think should be mentioned and possibly compared to as well.
3. What are the additional pre-processing time/space requirements in solving for low-rank compression? Some analysis of this would be good, and further comparing the tradeoffs between performance, pre-processing time, and inference time/space.
4. Tables should include some bolding or highlighting otherwise it's hard to tell what the reader is supposed to look at.
5. Figure and table captions can be made more informative and provide standalone arguments.

================Post rebuttal================

I thank the work done by the authors during the rebuttal. However, my main concerns regarding novelty and experimental comparisons still stand - I think it is insufficient to say that your method is 'complementary to most if not all existing methods, so we could easily combine the proposed method to the others to further accelerate' without showing this or comparing to current approaches. The other reviewers have also brought up important concerns regarding experimental details which I concur with.

---

> ### Author Response · Authors · 2020-11-22
> **Thanks reviewer for some useful comments. We believe the speedup is prominent and contribution is there.**
>
> 1. We thank for reviewer pointing out the need for Figure 3. We intended to demonstrate that there is a efficiency-efficacy tradeoff to justify that we select 3% accuracy loss as the anchor point. We have moved it to the Supplmentary.
>
> 2. We thank the reviewer for pointing out some other methods. As we illustrated in the paper, our method is complementary to most if not all existing methods, so we could easily combine the proposed method to the others to further accelerate. We will be adding these to the citation but in terms of empirical results we believe the current demonstration is still competitive to be considered as a major contribution.
>
> 3. This work focuses on the inference time acceleration. For most of real-world applications, accelerated inference methods would be applied many times in the production line. Thus, despite preprocessing time could cost some time, in general it’s an one time process and won’t affect too much as tons of millions of data would be processed in the subsequent usage. Previous work doesn't report the pre-processing time/space in general as well. For our method, it depends on the training data size so a smaller dataset such as STS-2 takes 10 mins and larger ones such as QQP takes hours. The same applies to the memory space as it relates to the data size. The whole procedure is on CPU and thus in general there is no memory issue.
>
> 4. We will be modifying the Table/Figures in the subsequent revisions.

---

### Official Review · AnonReviewer4 · 2020-10-30
**Limited contribution**

**Rating:** 5
**Confidence:** 4

**Review:**

The goal of this paper is to accelerate large-scale NLP models. This paper reduces the computational complexity by exploiting the data distribution. They claim that exploiting the data distribution enables us to perform low-rank approximation on feed-forward networks. Furthermore, they use that idea to reduce the complexity of the dot-product of attention modules. They experimentally show that they achieve faster inference time while retaining original accuracy. In addition, they show that their method can be combined with distillation methods.

Strength
- This paper is well organized. We easily understand the goal, challenges, and main idea. In addition, they provide an example of their observation which leads to their main idea.
- They achieve faster inference time with similar accuracy to a given BERT model; they combine low-rank approximation with a distillation method. They show the experimental results of BERT models as well as LSTM models.


Weakness
- Why not report the preprocessing time? It would not be a fair comparison to compare only the inference time.
- The accuracy loss seems to be non-negligible.
- In the last line of proof of Theorem 1, how can we get the right equation from the left equation? U and V are column-orthogonal matrices (not orthogonal matrix) so that UU^T and VV^T are not equal to identity matrices. If Theorem 1 is incomplete, need to re-investigate the method and experiments.
- They sub-sample 10% of the training data to perform low-rank approximation. The ratio affects the performance of a compressed model. I would like to see the performance change with respect to the sampling ratio.

---

> ### Author Response · Authors · 2020-11-22
> **Address the concerns in the reply in particular the derivation. The proposed method is technically sound.**
>
> 1. This work focuses on the inference time acceleration. For most of real-world applications, accelerated inference methods would be applied many times in the production line. Thus, despite preprocessing time could cost some time, in general it’s an one time process and won’t affect too much as tons of millions of data would be processed in the subsequent usage. Previous work doesn't report the pre-processing time in general as well. For our method, it depends on the training data size so a smaller dataset such as STS-2 takes 10 mins and larger ones such as QQP takes hours.
>
> 2. As shown in the paper that the accuracy loss is a trade-off between efficiency and efficacy. It’s faced by any approximation method and we could select a compression ratio causing less than 0.1% accuracy but still accelerate. Previous literature all needs to consider a balance between and in general people report the model with less than 3% accuracy loss to compare. It applies to all previous approximation methods to achieve a fair comparison. And indeed in the experiments we showed that DRONE achieved very good results and it can be combined with other methods, which is a significant contribution.
>
> 3. Indeed, multiplying V on the left hand side  of S_{W,r}^{-1} Z_k won’t give us the definition of M*. However, if you substitute M* back into the left hand side it suffices to see that such M* will give us Z_{k} and thus it obtains the optimal. This reflects the fact that M* is not an unique solution as we don’t have a one-to-one transformation from left-hand side to right. M* is only one of many solutions such that by using the derived formula, it could achieve the optimal decomposition error. And we only need to have one optimal solution in order to complete the proposed data-aware low-rank approximation.
>
> 4. Provided enough computational resources, we don’t need to sub-sample any data to compute a complete approximation. On SST-2, we have tried 5%,10%,20%,50% and 100% data used and the results are similar. It might be task-dependent but in general we found that 10% suffices.

---

### Official Review · AnonReviewer1 · 2020-10-31
**interesting idea but has experimental issues**

**Rating:** 3
**Confidence:** 3

**Review:**

This paper proposes a method to compress large-scale neural language models (e.g., BERT) without losing too much downstream performance. In contrast to prior work that seeks to compress just the parameters W in the standard linear layer h = Wx + b, this work incorporates the input by approximating Wx instead. The authors also extend this idea to approximations of attention, which enables them to experiment with Transformer architectures. Experiments across the GLUE benchmark confirm that the method does not significantly harm downstream performance, and it also seems to result in inference speedups. However, the biggest flaw with this paper is that its timing experiments are reported on a single CPU thread, while the majority of practical scenarios rely on GPUs. There are other issues with the choices made for the timing experiments (large batch size despite using CPU?) which make me skeptical that the method works as well as claimed.  As such, I cannot recommend the paper's acceptance.

comments:
- i'm not quite sure how the proposed method is more "generic" than existing related work. why does having to "train the proposed structure from scratch" (pg 1) reduce the practicality of a method? in general, these types of applications prioritize inference speedups over training time, so i don't get the criticism. i feel like touting the method as "generic" is a little misleading; the proposed method here is complementary to prior work on compression / distillation and should be presented as such.
- the related work comparisons are similarly exaggerated / misleading (sec 2). implying that pruning methods "might not" reduce inference speed is strange given that much prior work on pruning BERT-like methods yields significant inference speedups.
- this method assumes that the data lies in a low-dimensional subspace, which might not be a valid assumption for many tasks / datasets. i would have liked to see more discussion of this, although the paper does evaluate over many different tasks.
- the GLUE speedup ratios are interesting; do they correlate to something like the complexity of a particular dataset?
- a HUGE concern with the timing experiments is that they were done on a single thread of a CPU, not a GPU. especially with a large batch size (100 is fairly large), this definitely disadvantages the CPU computation for larger models. DRONE is not very relevant if the same speedups don't occur on a GPU; no one is serving BERT models on CPU. this makes me skeptical of the value of the proposed method, and is by itself enough to reject the paper.
- it's even more strange because "DRONE-retrain" performs 1 epoch of fine-tuning, presumably on a GPU, so why would a CPU be used for timing?

---

> ### Author Response · Authors · 2020-11-22
> **Response to the comments. GPU results are added and the speedup is strongly justified.**
>
> 1. We want to clarify that despite using a batch of size 100, we limit the thread to be 1 thus the inference time of batch size 1 is roughly the same as the ones listed in the tables/figures. When the thread is not limited to 1, some acceleration such as BLAS and MKL would come into work and the overall time for all methods will be improved. Specifically, all methods are about 8 times faster. Take STS-B for example, the original BERT improved from 420 ms to 50 ms and the proposed method improved from 29.5 ms to 4.1 ms. There is still over 10x faster on multi-thread environment.  We use this setup as this is a common scenario when the inference is done on devices (e.g., mobile phone or IoT edge devices). In these scenarios, number of cores is rather limited and multi-thread could still be costly. Since DRONE method itself is a dimension reduction scheme, a smaller model will certainly lead to a better acceleration.  We chose an evaluation setup such that the speedup ratio could genuinely reflect the change of model sizes. The GPU results are updated in the revised Table 4. Basically it still could achieve a 10x acceleration on various GLUE tasks. When the case is GPU, batch size = 1 could be harmful as GPU’s power would be exploited when the same instruction is operated on multiple instance of data. We need to collect at least 10 points to achieve the reported results on GPU. When the inference is on device, it’s usually the case as reviewer pointed out that the inference is done one-by-one. But when it’s done on GPU server in industry facilities, usually the service would collect multiple queries together to make the batch size larger in order to better leverage GPU. In sum, the acceleration speedup ratio is hardware dependent but overall the speedup trend is the same as DRONE could indeed reduce the model size to achieve the acceleration.
>
> 2. Most of the existing works of acceleration of transformer focus on accelerating the attention module. The “generic” term we refer here is that the technique could be applied to different types of modules (FFN, attention) in the model, and it’s indeed complementary to other types of methods. We will rephrase in the later version to make this clear.
>
> 3. We thank for reviewer pointing out that our writeups could mislead that pruning methods "might not" reduce inference speed. Our major claim should be that the reduction of model size in pruning methods might not have a corresponding effect of speed-up. That is, our own experiment showed that a 50% pruning will not lead to a near 2x inference time acceleration even as in low-rank methods. This finding is also found in other resources [1]. In the section “How much can we prune BERT, and what about acceleration?”, authors pointed out that 60% pruning will only lead to 28% increase in speed, which is rather limited compared to other types of model reduction methods. We understand the quantization could lead to great acceleration with hardware support [2] and it will be really helpful if the reviewer 1 could point us to the work where quantization with reported inference time on transformers.
>
> 4. The GLUE complexity could be reflecting to the number of training data points each task has. Essentially there are some large datasets such as QQP and other smaller ones such as SST-2.
>
> 5. Low-dimensional assumption is an empirical assumption which is also observed in previous NLP literature. We have empirically validated the assumption by showing spurious performance on various dataset. Notice that under the worst case that such assumption does not hold, X doesn’t represent any information ant thus it degenerate to an Identity Matrix and the original problem becomes the solving SVD. This means that the proposed DRONE is a strictly better method than SVD low-rank approximation.
>
> [1] https://blog.rasa.com/pruning-bert-to-accelerate-inference/
> [2] https://dl.acm.org/doi/pdf/10.1145/3007787.3001163

---

### Decision · Program_Chairs · 2021-01-07
**Final Decision**

**Decision:**

Reject

**Comment:**

This paper proposes a method for compressing weight matrices in large scale pre-trained NLP encoders (like BERT) through low-rank decompositions of both fully connected and self-attention layers. The method is used to compress and speedup pre-trained models. Experiments measure timing on a single CPU thread and demonstrate speedups with small loss of accuracy. Reviewers noted the that goal of this paper is potentially impactful. Some reviewers viewed the resulting loss in accuracy as marginal, while others viewed it as more substantial -- a potential downside. Reviewers also raised concerns about the methodology used to measure inference speedup, a critical measure of success. Specifically, timing experiments were done only on a single CPU thread -- while most practical scenarios would almost certainly rely on GPUs -- as a result, positive experimental results are less impactful. Authors updated the paper to include GPU timing experiments, which did show speedups -- though only marginal speedups over the baseline, TinyBERT. Further, reviewers pointed out that there are several other relevant baselines on compression approaches that are not compared with, and that further analysis should be done on the timing/accruacy tradeoff of baseline methods. Finally, reviewers felt that the contribution of the proposed method relative to other approaches that also attempt to compress transformers is not clearly outlined. Weighing these concerns, I agree with reviewers that the paper is not ready for acceptance in its current form.